# Quantifying the Performances of the Semi-Distributed Hydrologic Model in Parallel Computing—A Case Study

**JungJin Kim [1] and Jae Hyeon Ryu [2],*** 

[1]   Texas A&M AgriLife Research (Texas A&M University System), P.O.Box 1658, Vernon, TX 76384, USA; jungjin.kim@ag.tamu.edu
[2]   Department of Soil and Water Systems, University of Idaho, 322E. Front ST, Boise, ID 83702, USA
*   Correspondence: jryu@uidaho.edu; Tel.: +1-(208)-332-4402

**Abstract:** The research features how parallel computing can advance hydrological performances associated with different calibration schemes (SCOs). The result shows that parallel computing can save up to 90% execution time, while achieving 81% simulation improvement. Basic statistics, including (1) index of agreement (D), (2) coefficient of determination ($R^2$), (3) root mean square error (RMSE), and (4) percentage of bias (PBIAS) are used to evaluate simulation performances after model calibration in computer parallelism. Once the best calibration scheme is selected, additional efforts are made to improve model performances at the selected calibration target points, while the Rescaled Adjusted Partial Sums (RAPS) is used to evaluate the trend in annual streamflow. The qualitative result of reducing execution time by 86% on average indicates that parallel computing is another avenue to advance hydrologic simulations in the urban-rural interface, such as the Boise River Watershed, Idaho. Therefore, this research will provide useful insights for hydrologists to design and set up their own hydrological modeling exercises using the cost-effective parallel computing described in this case study.

**Keywords:** hydrologic simulation; HSPF; BEOBEST; parallel computing; Boise River Watershed

---

## 1. Introduction

A hydrologic model is commonly used to simulate real-world problems in many water-related fields, including hydrological, ecological, biological, and environmental studies [1–4]. Recent advances in data-intensive products, such as North American Land Data Assimilation system (NLDAS) and NEXt Generation RADar (NEXRAD) enable hydrologists to better characterize hydrological processes at higher spatial and temporal scales [5–8]. However, it is a daunting task for hydrologists to calibrate their models using these data-intensive inputs.

Due to insufficiently observed datasets, the computer simulation approach is a typical exercise to characterize hydrological processes and to enhance hydrological simulations based on physical and conceptual parameters. In general, hydrologists utilize the selected key parameters to calibrate their models for efficient simulations associated with cost and time [9–14]. However, model performances are constrained by the number of sets of parameters used, which do not necessarily ensure that the selected model performs best to characterize hydrological processes in a complex watershed. Therefore, computer parallelism is a way to enhance simulation performances when many parameters are considered for further adjustments in hydrological modeling settings. Thanks to computer parallelism, computational modeling has rapidly advanced [15–17]. Although computer speed and capacity improve over time, the model calibration time is still challenging for many practitioners [18].

There are two typical approaches to parallelize hydrological simulations. First, a parallel algorithm with parallel threads (e.g., multiple cores) within a single computer is one approach [19–23]. The other approach is implementation of a parallel algorithm in connection with multiple machines [24,25]. Although several studies have been conducted for parallelizing model calibrations to reduce execution time and effort using multiple threads in a single machine [26–28], few studies focus on quantifying how multiple machines associated with cluster-based computing architecture can improve model performances in the field of hydrology. Moreover, computer parallelism on a cluster-based framework has not been fully implemented to find optimal parameters for hydrological simulations, especially Hydrological Simulation Program–Fortran (HSPF) modeling settings. Therefore, this research explores how computer parallelism can be implemented to evaluate the enhancement of hydrological simulations using HSPF so that hydrologists can apply it to their own applications.

Figure 1 shows a flowchart of computer parallelism to calibrate HSPF in a Linux cluster framework. A small Linux cluster system (sLCS) is first developed along with one master and six slave nodes. Next, climate data and geographical information are used to create sub-watersheds using a built-in delineation tool in BASINS 4 Software [29], and then climate data are routed into HSPF to generate streamflow. Once the simulated streamflow is generated by HSPF, calibration and validation exercises in computer parallelism are conducted to evaluate how well HSPF performs characterization of hydrological consequences associated with climate and land-use/land-cover (LULC) profiles in the study area.

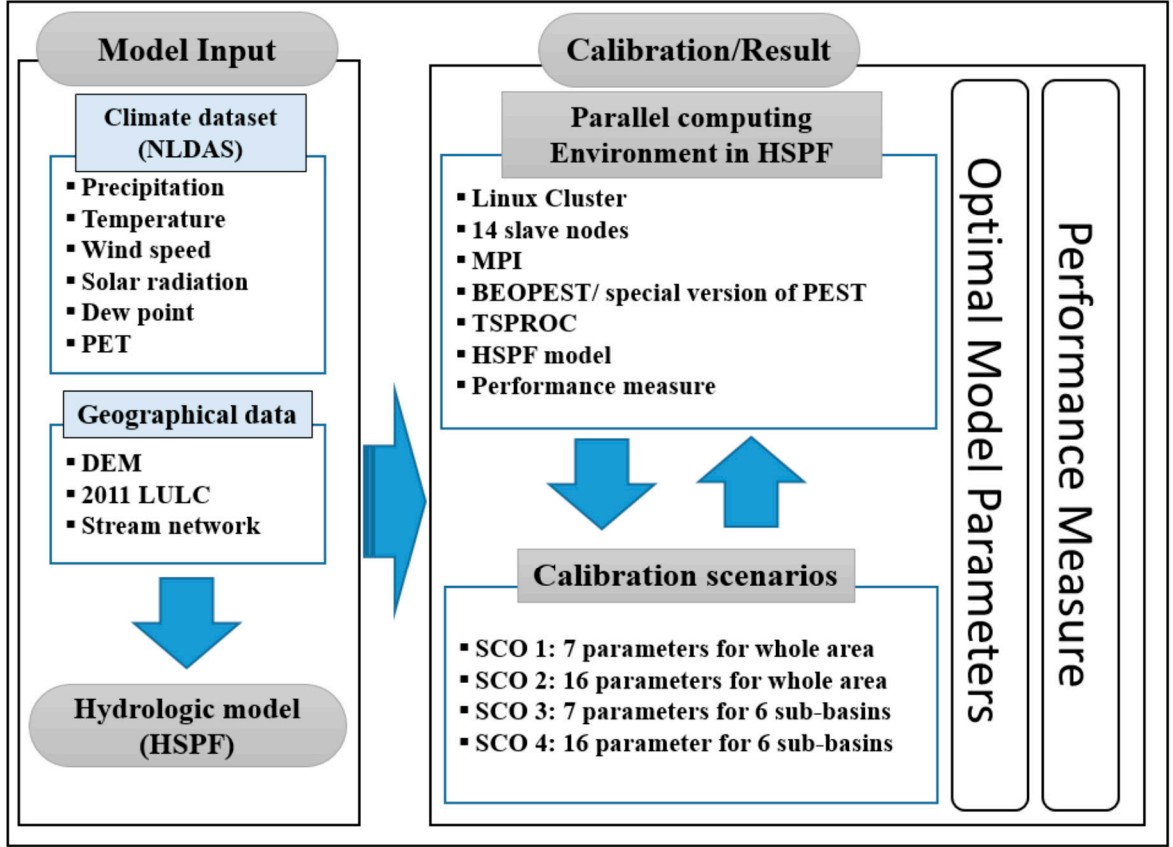

**Figure 1.** The procedual diagram of Hydological Simulation Program–Fortran (HSPF) simulations in parallel computing.

Four different calibration schemes (described later) are used to determine optimal calibration scenarios in computer parallelism. For example, the BEOPEST, a special version of PEST (a model-independent parameter optimization program) [30] is used as a tool to calibrate HSPF with

14 cores using a message passing interface (MPI) in sLCS and all simulation outcomes associated with such schemes are then evaluated based on performance criteria as listed in the Appendix A. The scheme includes: (1) index of agreement (D), (2) coefficient of determination ($R^2$), (3) root mean square error (RMSE), and (4) percentage of bias (PBIAS). Once the best calibration scheme is selected, additional efforts are made to improve model performances at the selected calibration target points, while the Rescaled Adjusted Partial Sums (RAPS) is used to evaluate the trend in annual streamflow. The result indicates that hydrologic simulations using BEOPEST and HSPF in sLCS environment is a way to improve model performances, especially when many parameters at the complex watershed are used for model calibration exercises.

## 2. Study Area and Data

The Boise River Watershed (BRW) is selected as the study area (Figure 2). As a tributary of the Snake River system, the BRW plays a key role of providing water to the Boise metropolitan areas, including Boise, Nampa, Meridian, and Caldwell. The drainage area of the basin is about 10,619 km$^2$ with a mainstream length of 164 km and flows into the Snake River near Parma. More than 40 percent of Idaho residents live in this basin and 60 percent of the people reside around the floodplain. The main physiographic characteristic of the BRW is that a greater proportion of precipitation falls as snow at higher elevations. It causes the predictability of high flows due to the snow melting process, and therefore a localized flood event is often observed during late spring and early summer.

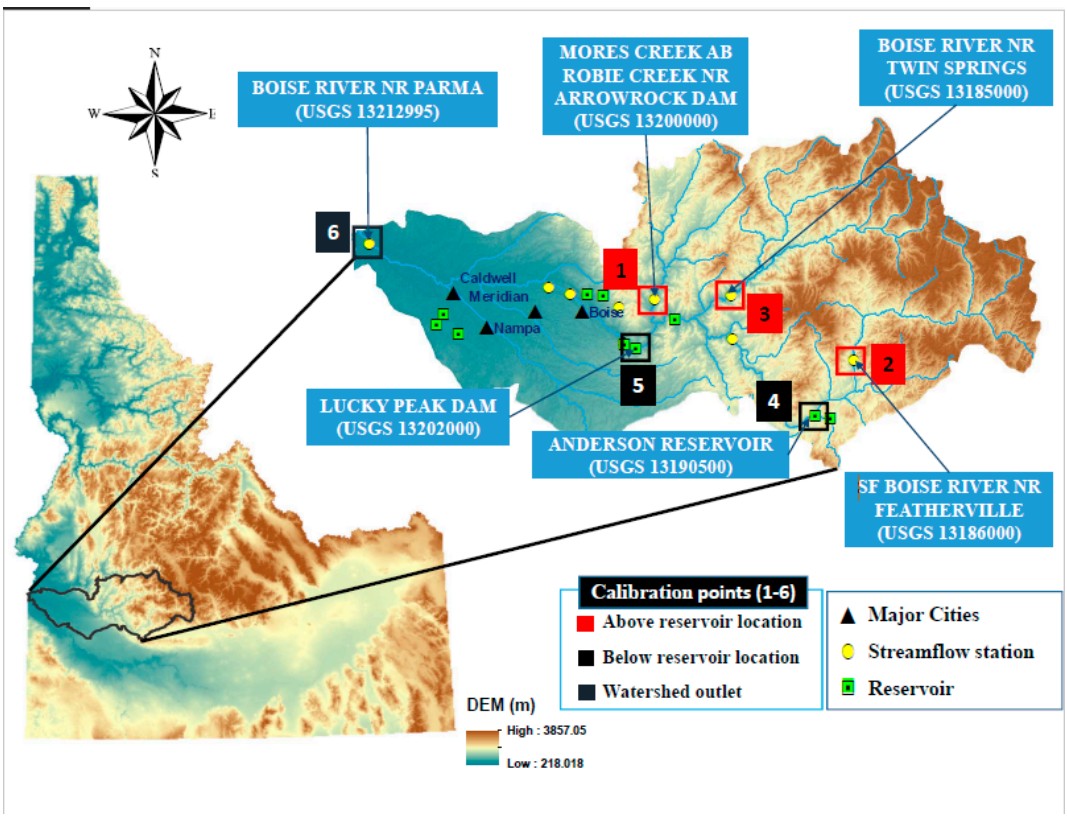

**Figure 2.** The study area, the Boise River Watershed adapted from [9].

To simulate streamflow using HSPF, it requires primary input datasets, including precipitation, temperature, and potential evapotranspiration (PET). Phase 2 of the North American Land Data Assimilation System (NLDAS-2) data were used as climate forcing because a series of required climate data (e.g., precipitation, temperature, downward solar radiation, downward long wave radiation, wind speed, specific humidity, surface pressure, potential evapotranspiration, and others) were available.

These datasets were in the eighth-degree grid spacing and used for the simulation period from 1 January 1979 to 31 December 2015 (36 years) at hourly time-steps. Note that the NLDAS-2 data have been examined along with the observed data product in several studies [31–35].

The derived climate data from NLDAS-2 were then converted to the watershed data management (WDM) format to be used as inputs for HSPF. However, there were a few issues with the conversion of the data from NLDAS-2 to WDM using conventional tools, which required significant time and effort for all 112 grid points at the BRW. Since the existing WDM utility tool could not import a large volume of forcing data (roughly about 30 MB per single file), a R script [36] was used to extract forcing data from NLDAS-2 to a WDM file. The SARA Time Series Utility [37] was then utilized to create a complete set of the WDM file.

A 30 m spatial resolution interval of digital elevation model (DEM) provided by the U.S. Geological Survey (USGS) was used to delineate watersheds and to determine flow directions in BASIN 4.1 modeling platforms [29]. The National Hydrography Dataset (NHD) and DEM were then used to characterize stream routing processes at a functional sub-watershed (1:100,000). A total of six observed streamflow stations were selected for calibration target points (TPs), including three points above reservoirs (no major diversion is found), two points below reservoirs, and one point at watershed outlet (see Figure 2). Additionally, land-use/land-cover data (LULC) in year 2011 [38] was used to classify land segments, such as urban, agricultural land, forest land, water/wet land, shrub land, grass land, and barren/mining land (Figure 3). Model calibration and validation effort were then made from 1 January 1999 to 31 December 2015 (17 years) and 1 January 1979 to 30 December 2000 (22 years), respectively. Note that there was a missing period from October 1, 1997 to December 2000 at the calibration target point six (TP6) so that the period from 1 January 1979 to 30 September 1997 was used for calibration.

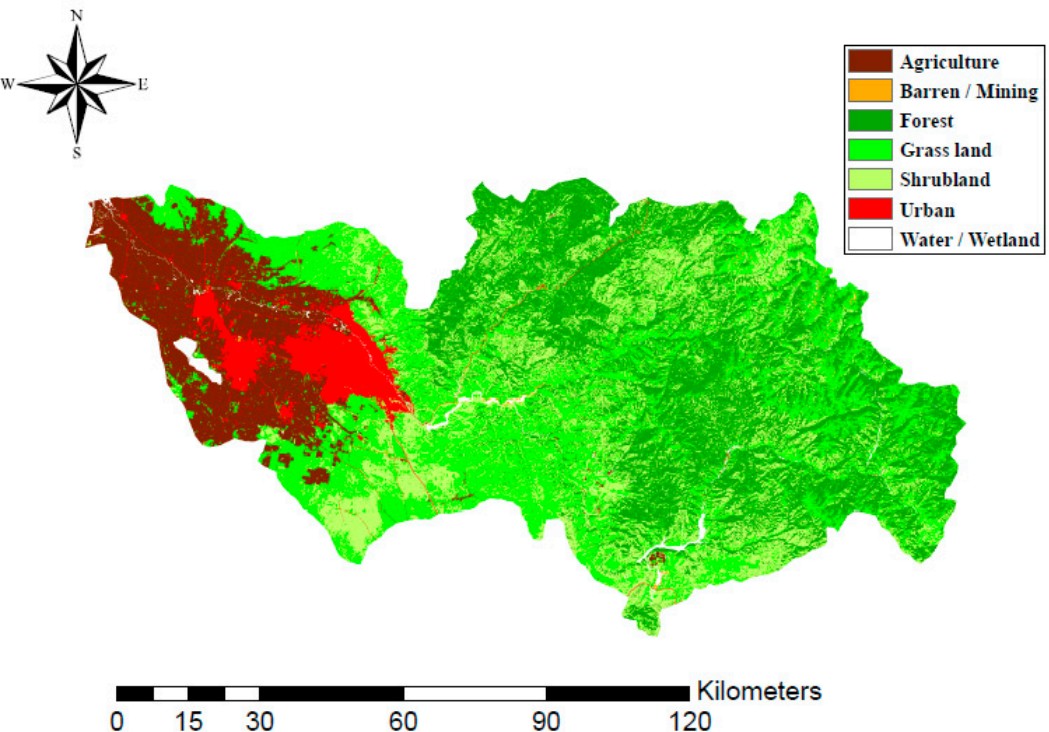

**Figure 3.** Land use and land cover at the study area, the Boise River Watershed.

## 3. Methodology

### 3.1. Small Linux Cluster System (LCS)

A small Linux cluster system (sLCS) was designed and built for this study using a multimode Beowulf, which is a portable computer cluster compatible with various computer architectures [39]. Beowulf is a local memory machine using messaging through local network linking master and slave nodes via local ethernet networks (TCP/IP) so that it can support open multi-processing (OpenMP) [40], message passing interface (MPI) [41], and compute unified device architecture (CUDA) parallelism [42]. A main advantage of sLCS is that it is easy to use and it is cost-effective to build high-performance computing for a small research group at a university and/or a small business since it costs less than $3000 (e.g., 6 × VIA CN10000 with 2 GHZ CPU, 1GB of RAM, 500GB of hard disk, 1Gbps Ethernet card). A typical sLCS is composed of 1 master and 6 slave nodes that are controlled by the master node and linked to each other via TCP/IP. For this study, 22 cores (8 cores in master node, 4 cores + 2 cores each × 5 slave nodes = total 22 cores) were used to implement parallelism during the model calibration. More specifically, a laptop was used as the master node, while the other slave nodes were connected to each other via TCP/IP as shown in Figure 4. The primary roles of the master node included: (1) to use resources for running software, (2) to exchange model parameters with the slave nodes, and (3) to save and display simulation results. Note that Ubuntu 64-bit version [43] was used as an operating system (OS).

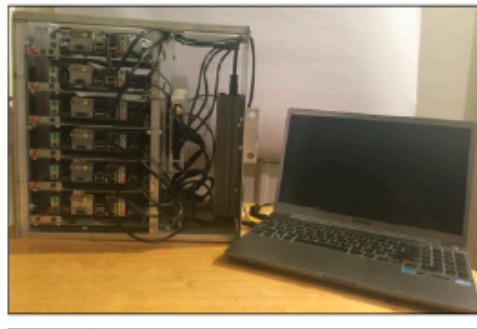

**Typical sLCS**

| Name | Content | Unit |
|---|---|---|
| OS | BCCD: Bootable Cluster CD 64 bit (Linux) | 1 |
| CPU | Intel Atom Dual Core (D525/510) | 6 |
| Core | Master node (4 cores) | 4 |
| | 5 slave nodes (2 cores) | 10 |
| Ram | 2 GB | 6 |
| Hard Drive | 2.5" SATA 160GB | 1 |
| | | |
| Software compatibility : Low | | |

**Modified sLCS**

| Name | Content | Unit |
|---|---|---|
| OS | Ubuntu 64 bit (Linux) | 1 |
| CPU | Intel i-7 2.3GHz | 1 |
| | Intel Atom Dual Core (D525/510) | 6 |
| Core | Master node (8 cores) | 8 |
| | 1 slave node (4 cores), 5 slave nodes (2 cores) | 14 |
| Ram | 8 Gb (laptop) | 1 |
| | 2 GB | 6 |
| Hard drive | 300 GB SSD | 1 |
| | | |
| Software compatibility: High | | |

**Figure 4.** System specification of the small Linux cluster system (sLCS).

*3.2. System Setup*

A diskless sLCS in Beowulf system was developed in Ubuntu Operating System (OS) [44]. Slave nodes with no hard disk were connected by an Ethernet network hub to the master node that could control, supervise, and monitor other salve nodes. The MPI library was used to coordinate multiple processes in a distributed memory environment. For communication protocols, the Secure Shell (SSH) method was used along with the Ubuntu-based diskless remote boot system (UDRB) to manage cluster nodes. Thus, the UDRB installed in the master node provided a diskless environment for the slave nodes, accessing local hardware. A wireless network (WiFi) was used for the master node to access the internet network, while all other connections between master and slave nodes were linked by network cards.

*3.3. Hydrologic Simulation Program–Fortran (HSPF)*

HSPF is a process-based, river basin-scale, and semi-distributed model for hydrologic simulations [45]. This model was used to simulate the impact of land management and/or climate change on water, sediment, and water quality in large and complex watersheds. In addition, HSPF was used to simulate water quality and quantity at various basin scales and locations (e.g., urban, agricultural, mountain area), where complicated water issues are intertwined between the states and/or the countries. HSPF consisted of three main modules (PERLND, IMPLND, and RCHRES) and an additional optional utility module. Each module had different state variables representing water quality and hydrological processes [45]. Further compiling effort is needed to make HSPF compatible with the Linux environment so that it can parallelize the calibration processes using BEOPEST in sLCS.

*3.4. Time-Series Processor (TSPROC)*

A tool known as a general time-series processor (TSPROC) is an interface to assist seamless data exchanges between input and output for optimal parameterizations in hydrologic simulations. Basically, TSPROC generated the key input file for the parameter estimation (PEST) program (which minimized model biases and errors of estimation formulated in a user-specified objective function). To fully implement TSPROC in sLCS, compilation of TSPROC was also required (the compilation process is not shown in the paper) because the current version of TSPROC was compiled for Windows only.

*3.5. BEO-Parameter Estimation (BEOPEST)*

PEST, the model-independent nonlinear parameter estimation and optimization tool developed by [30] was used to assist with data interpretation, model calibration, and predictive analysis. PEST used a recursive gradient-based optimization technique, linearizing the nonlinear problem by iteratively computing the Jacobian matrix of sensitivities of model observations to parameters. The parameter estimation in PEST was accomplished using the Gauss–Marquardt–Levenberg algorithm (GML) to minimize the user-defined objective function (e.g., minimization of root mean squares between simulated and observed values). The BEOPEST was a tool to mitigate the computation burden and implement parallelism in PEST [46]. The BEOPEST was installed in the master node and it communicated with the slaves without any additional physical file exchanges. Thus, two communication protocols, such as Transmission Control Protocol/Internet protocol (TCP/IP) and MPI were commonly used. Therefore, throughout TCP/IP, the BEOPEST and MPI were utilized to run HSPF through data exchanges in a diskless Linux cluster environment, such as sLCS. As library sources, an OPENMPI library was installed to compile a parallel code fully workable in sLCS. Since BEOPEST in sLCS was a cost-effective approach and powerful, it was highly recommended to execute model calibration in computer parallelism with affordable costs for a small research group.

*3.6. Streamflow Calibration Schemes*

BASINS Technical Note 6 [47] provided guidance on hydrologic and hydraulic parameters including parameter definition, the units, and acceptable ranges for HSPF. Table 1 lists parameter name, unit, initial value, and ranges for streamflow calibrations. Four different calibration schemes were used for the simulation period, and the first two years (1 January 1999 to 31 December 2000) were selected as a warm-up period to reduce the sensitivity of the model results to the assumed initial conditions (see Table 1).

**Table 1.** Initial values and range of values for streamflow parameters for the HSPF model.

| Parameter | Definition | Units | Initial Value | Range of Values | |
|---|---|---|---|---|---|
| | | | | Typical [1] | Possible [1,2] |
| AGWETP | Fraction of remaining potential evapotranspiration from active groundwater | None | 0 | 0.0–0.05 | 0–1.0 |
| *AGWRC** | *Base groundwater recession rate* | *None* | *0.98* | *0.92–0.99* | *0.82–0.999* |
| *BASETP** | *Fraction of potential evapotranspiration from baseflow* | *None* | *0.02* | *0.0–0.05* | *0–1.0* |
| CEPSC | Interception storage capacity | mm | 2.54 | 0.76–5.08 | 0.25–254 |
| DEEPFR | Fraction of groundwater inflow to deep recharge | None | 0.1 | 0.0–0.2 | 0.0–1.0 |
| *INFILT** | *Infiltration rate* | *mm/hour* | *4.06* | *0.25–6.35* | *0.03–12.70* |
| INTFW | Interflow inflow parameter | None | 2.0 | 1.0–3.0 | 0.0–10.0 |
| *IRC** | *Interflow recession parameter* | *1/day* | *0.5* | *0.5–0.7* | *0.1–0.9* |
| KVARY | Variable groundwater recession flow | 1/mm | 0 | 0.0–76.2 | 0.0–127.0 |
| LZETP | Lower zone evapotranspiration parameter | None | 0 | 0.0–0.7 | 0.1–0.9 |
| LSUR | Length of the assumed overland flow | m | 152.4 | −60.96–152.4 | 30.48–304.8 |
| *LZSN** | *Lower zone nominal soil moisture storage* | *mm* | *152.4, 165.1* | *76.2–203.2* | *50.8–381.0* |
| NSUR | Manning's roughness for overland flow | None | 0.2 | 0.03–0.1 | 0.01–1.0 |
| *SLSUR** | *Slope of overland flow plane* | *None* | *0.001* | *0.30–1.52* | *0.0001–304.8* |
| *UZSN** | *Upper zone nominal soil moisture storage* | *mm* | *28.7* | *2.54–25.40* | *0.25–254.0* |
| INFEXP | Exponent in infiltration equation | none | 2.0 | 2.0–2.0 | 1.0–3.0 |

* indicates the model parameters that were used by [9]. [1] BASINS Technical Note 6 [47]. [2] HSPF Version 12.4 User's Manual [48]).

Specifically, schemes (SCOs) 1 and 2 were designed to calibrate whole basin with different parameter sets. Thus, scheme 1 (SCO1) used 7 model parameters provided by [9] because these parameters are commonly used for model calibration regardless of watersheds. Scheme 2 (SCO2) used 16 model parameters (see Table 1), while scheme 3 (SCO3) and scheme 4 (SCO4) are designed to calibrate the model with different parameter sets for 6 independent sub-watersheds shown in Figure 5. Thus, SCO3 uses 7 model parameters for 6 sub-watersheds (total 42 model parameters = standard 7 model parameters × 6 sub-watersheds) and SCO4 uses 16 model parameters for 6 sub-watersheds (total 96 model parameters = 16 model parameters × 6 sub-watersheds). For each calibration scheme, computer parallelism was applied to evaluate its performances using BEOPEST in sLCS.

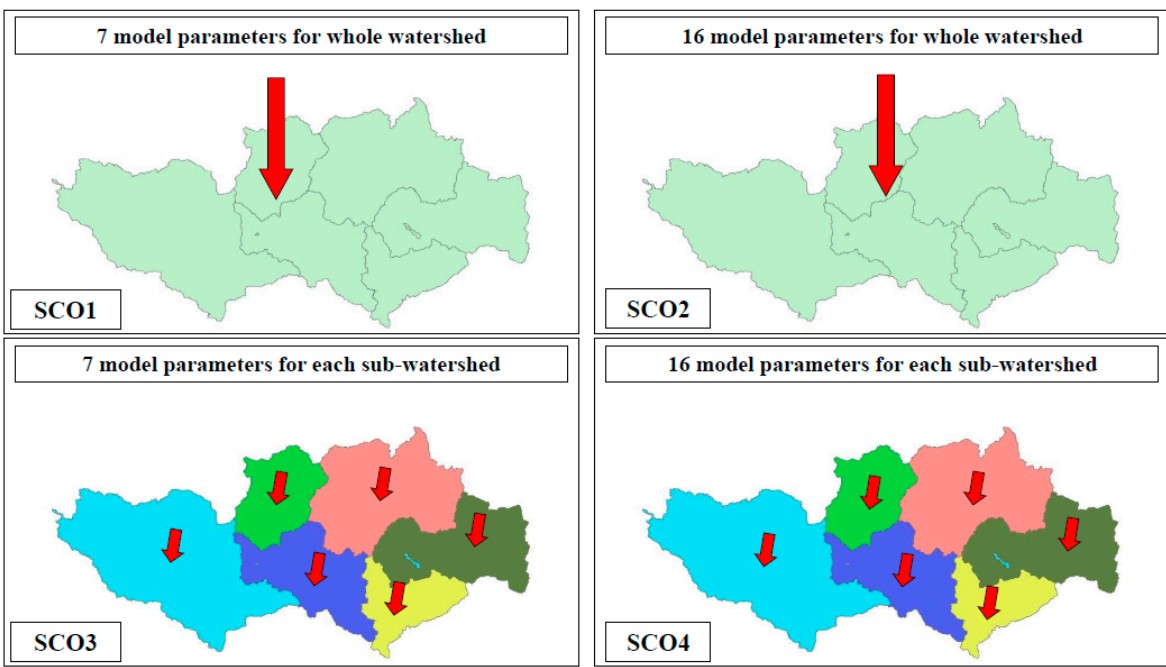

**Figure 5.** Four different calibration schemes (SCOs) in multiple sub-watersheds.

*3.7. Performance Measures*

### 3.7.1. Performance Measures for Parallel Computing

To evaluate parallel performances, various evaluation criteria, including run time, time reduction, speedup, efficiency, scalability, and more were considered, however, program run time (PT), percentage of time reduction (PP), parallel speedup (PS), and parallel efficiency (PE) were selected for the sake of convenience. Parallel speedup (PS) was defined as the degree of true time reduction between a serial computation and parallel computation, and this measure indicated the relative improvement of model performance during calibrations. A notation of PS is proposed by Amdahl's law [49] and it was used to compute the theoretical maximum parallel speedup, when multiple processors were used. It was denoted as:

$$\text{PS} = \frac{T_s}{T_P} \tag{1}$$

where, $T_s$ = execution time of a serial computation on a single process core, *s*. $T_P$ = execution time of parallel application on multiple processors, *p*.

Parallel efficiency (PE) is another way to measure the effectiveness of multiple processors. Under an ideal condition in computer parallelism, PE should be equal to all the cores used with maximum efficiency, which is 1. Although PE varied depending upon the number of cores used, PE should be between 0 and 1 in real-world applications due to the interference of physical components associated with load balancing, lack of hardware capacity, network connection, and other physical constraints, if any. PE was denoted as:

$$\text{PE} = \frac{S}{P} \tag{2}$$

where, *S* = efficiency of a single process core, *P* = efficiency of multiple processors.

### 3.7.2. Performance Measures for HSPF Simulations

Six typical performance measures, including index of agreement (D), coefficient of determination ($R^2$), Nash–Sutcliffe efficiency (NSE), root mean square error (RMSE), RMSE-observations standard deviation ration (RSR), and percentage of bias (PBIAS), were selected to evaluate how well HSPF simulated streamflow as compared with the observed streamflows at the BRW. D was the insensitivity

of the correlation-based measure to variances [50]. It ranged from 0.0 to 1.0. $R^2$ was the degree of collinearity between the observed and simulated values. It ranged from 0.0 to 1.0. Note that higher values of D and $R^2$ indicated better agreement between the simulated data and the observed data. Typically, if the $R^2$ value was greater than 0.5, acceptable model performances were granted [51,52]. The NSE was the percentage of the observed variance explained by the model and determined the efficiency criterion for the model verification [53]. It ranged from minus infinity to 1.0, with higher values indicating better agreement between the observed data and the simulated data. If the NSE value was greater than zero, the model was deemed a better system simulation than that of the mean of the observed data. The RMSE was an absolute error measure, quantifying error with regards to the variable units. It calculated a measure of the difference between the simulated data and the observed data. The individual differences were called residuals. The RMSE aggregated them into a single measure of predictive power. A lower value of RMSE showed better model performance and zero value indicated a perfect fit. The RSR was a standardized RMSE using the observed standard deviation. It incorporated both an error index and the additional information recommended by [54]. The lower RSR value (e.g., close to zero) indicated better model performance. The PBIAS was calculated to determine the average tendency of the simulated values as larger or smaller than observed counterparts [55]. A value of zero was the optimal model performance. Positive values indicated the underestimated bias, while negative values indicated the overestimated bias for the simulated results against the observed values.

### 3.7.3. Streamflow Analysis in Time Series.

The Rescaled Adjusted Partial Sums (RAPS) method [56] was used to detect and quantity trends fluctuation of simulated streamflow at the watershed outlet. This method overcame small systematic changes and variability in the time series. Note that trend, data clustering, irregular fluctuations, and periodicities in the time series can be represented by the RAPS visualization. The RAPS was calculated using the equation below:

$$RAPS_k = \sum_{t=1}^{k} \frac{Y_t - \overline{Y}}{S_Y} \tag{3}$$

where, $\overline{Y}$ is the mean data for entire data, $S_Y$ is standard deviation over the entire data, $k$ ($k$ = 1, 2, 3, 4, ..., $n$) is the counter limit of the summation for $k$-th year, and n is the number of the values in the time series.

## 4. Results

### 4.1. Parallel Performance

Parallel performances in sLCS are evaluated based on four different calibration schemes (SCO1–SCO4) using BEOPEST. Figure 6 shows the results of total program (calibration) run time (PT), percentage of time reduction (PP), parallel speedup (PS), and parallel efficiency (PE) by the number of core processes with respect to SCOs. Obviously, PT decreases as the number of cores increases. Note that PT of SCO4 is about ten times longer than that of SCO1 when a single core is used with the seven parameters (not shown in this paper). However, when model calibrations are conducted using two to eight cores, PT gradually decreases until no distinct improvement is observed at nine cores and above. The PP also shows a similar pattern in the sense that calibration with multiple cores can have time saving advantages. Therefore, the reduction rate of the total calibration time (PP) is achieved for 76%, 89%, 89%, and 90% from SCO1, SCO2, SCO3, and SCO4, respectively.

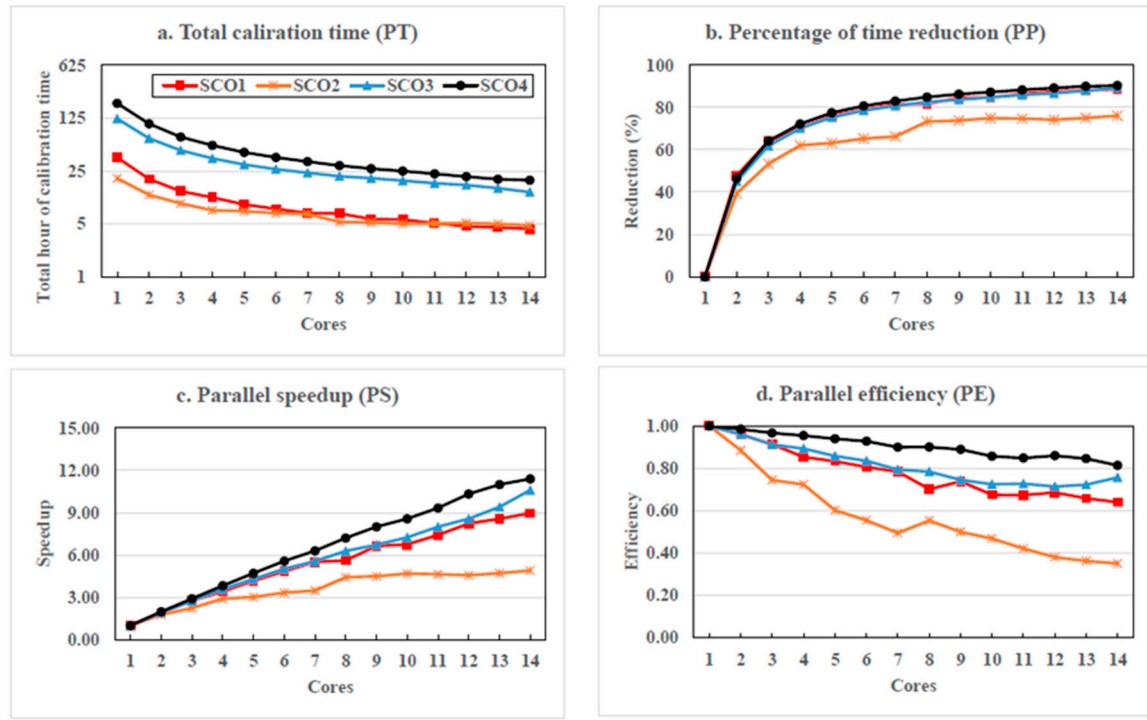

**Figure 6.** Parallel performance measures, including (**a**) total calibraiton time (PT), (**b**) percentage of time reduction (PP), (**c**) parallel speedup (PS), and (**d**) parallel efficiency (PE) in sLCS environment.

Based on the values of PS after calibration in parallelism, SCO1 do not gain many benefits as compared with that of the other schemes. Thus, the values of PS from SCO2 and SCO3 gradually increase as the number of cores increases, while that from SCO4 is the highest. It implies that the loss of speedup is due to the communication overhead when more processor nodes are added to the sLCS. Theoretically, if the number of parallel jobs is set, each core simultaneously reads files to be written in hard disk via TCP/IP. For this reason, the speedup will not reach 14, even if 14 cores are fully used due to network constraints. Similarly, PE is most likely a less-than-ideal value, which is one, because of system overhead issues associated with physical constraints (e.g., network bandwidth and/or throughput between cores). The results show that SCO2 has the lowest PE, and SCO4 has the highest PE. Overall, SCO4 has the best parallel performance as compared with other calibration schemes, regardless of number of cores. This implies that BEOPEST in sLCS settings works well, especially when many hydrological parameters need to be calibrated at multiple sub-watersheds.

### 4.2. HSPF Model Performance

In addition to the computer parallelism aspect, HSPF performances are also observed to evaluate how streamflow simulations are well made, associated with the historical data at the selected calibration target points (PTs). Table 2 shows the comparison of the model performance criteria for SCO1–SCO4. SCO1 and SCO2 are first compared to see how the different number of parameters affect the model performance. The result shows that $R^2$, d, NSE, and PBIAS of SCO 2 are higher than that of SCO1, but RMSE and RSR of SCO 2 are lower than SCO 1. It seems that SCO2 is more affected by the volume variation of streamflow driven by using more parameters. Obviously, HSPF performances after calibration improve against the no calibration option based on the performance criteria (see Table 2). Overall, SCO4 is the best, with higher NSE and D, and lower RSR, RMSE, and PBIAS, as compared with any other schemes, including the no-calibration option. As such, SCO4 is selected for additional effort to calibrate the interior calibration target points (TP1–TP5) at the BRW.

**Table 2.** The model performance comparison of calibrated monthly streamflow by calibration schemes at the calibration target point 6 (watershed outlet).

| Criteria | OBS | No Cal | SCO1 | SCO2 | SCO3 | SCO4 |
|---|---|---|---|---|---|---|
| Daily mean streamflow (m$^3$/sec) | 30.95 | 27.01 | 24.21 | 24.04 | 25.55 | 27.44 |
| D | - | 0.66 | 0.50 | 0.60 | 0.54 | 0.74 |
| R$^2$ | - | 0.30 | 0.41 | 0.45 | 0.41 | 0.37 |
| NSE | - | 0.17 | 0.24 | 0.31 | 0.28 | 0.34 |
| RMSE (m$^3$/sec) | - | 28.87 | 27.64 | 26.30 | 26.89 | 25.76 |
| RSR | - | 0.91 | 0.87 | 0.83 | 0.85 | 0.81 |
| PBIAS (%) | - | 12.74 | 21.78 | 22.32 | 17.46 | 11.33 |

*4.3. Results of Calibrated and Validated Streamflow Using SCO4*

SCO4 is now employed to calibrate all six TPs and Table 3 shows the final set of calibrated model parameter values at each calibration target point. Note that the same model parameters are initially assigned to six sub-watersheds, but the optimal parameter values differ from each of the others after calibration. Table 4 lists the statistical results after model calibration and validation using SCO4 at all six TPs.

**Table 3.** The final set of calibrated model parameters at six calibration target points.

| Parameter | Units | Calibration Target Points (TPs) | | | | | |
|---|---|---|---|---|---|---|---|
| | | TP1 | TP2 | TP3 | TP4 | TP5 | TP6 |
| AGWETP | None | 0.0001 | 0.0004 | 0.0003 | 0.0001 | 0.0126 | 0.0025 |
| AGWRC | None | 0.9796 | 0.9813 | 0.9808 | 0.9164 | 0.9782 | 0.9401 |
| BASETP | None | 0.2256 | 0.0864 | 0.1000 | 0.0362 | 0.1867 | 0.2000 |
| CEPSC | mm | 12.70 | 12.70 | 2.58 | 12.70 | 8.51 | 12.70 |
| DEEPFR | None | 0.5000 | 0.5000 | 0.1016 | 0.5000 | 0.3352 | 0.5000 |
| INFILT | mm/hour | 1.58 | 4.49 | 2.08 | 0.03 | 4.41 | 1.38 |
| INTFW | None | 0.7209 | 0.4337 | 0.8227 | 0.1173 | 0.4422 | 0.1932 |
| IRC | 1/day | 0.9000 | 0.8562 | 0.9000 | 0.9000 | 0.8569 | 0.4687 |
| KVARY | 1/mm | 0.00 | 0.00 | 0.00 | 0.00 | 0.00 | 3.72 |
| LZETP | None | 0.5000 | 0.5264 | 0.9648 | 0.2698 | 0.1000 | 0.1000 |
| LSUR | m | 12.05 | 35.05 | 28.81 | 238.91 | 64.05 | 243.84 |
| LZSN | mm | 50.80, 58.13 | 50.80, 158.23 | 50.8, 242.73 | 68.45, 63.62 | 56.81, 156.96 | 50.80, 71.39 |
| NSUR | None | 0.1317 | 0.1533 | 0.1260 | 0.9989 | 0.2825 | 0.0218 |
| SLSUR | None | 4.5068 | 0.2193 | 0.3623 | 0.0033 | 1.2819 | 0.1635 |
| UZSN | mm | 65.99 | 5.34 | 2.54 | 195.64 | 2.68 | 62.73 |
| INFEXP | None | 2.00 | 2.00 | 2.00 | 2.00 | 2.0 | 1.00 |

**Table 4.** Statistical results after calibration and validation using SCO4 at the six calibration target points (TP1–TP6) in the BRW.

| Calibration Target Point | | Daily Mean Streamflow (m³/s) | Evaluation Statistic | | | | | |
|---|---|---|---|---|---|---|---|---|
| | | | R² | D | RSR | NSE | RMSE | PBIAS (%) |
| TP1 | Cal | 5.02 (5.90) | 0.82 | 0.95 | 0.45 | 0.79 | 3.44 | 14.88 |
| | Val | 7.40 (8.09) | 0.76 | 0.92 | 0.60 | 0.64 | 6.07 | 8.60 |
| TP2 | Cal | 15.15 (17.04) | 0.85 | 0.93 | 0.41 | 0.83 | 8.43 | 11.09 |
| | Val | 17.65 (21.27) | 0.78 | 0.93 | 0.49 | 0.76 | 13.31 | 17.03 |
| TP3 | Cal | 32.63 (30.95) | 0.84 | 0.95 | 0.41 | 0.83 | 13.57 | 6.83 |
| | Val | 35.40 (34.41) | 0.82 | 0.95 | 0.45 | 0.80 | 16.50 | −2.88 |
| TP4 | Cal | 22.75 (21.56) | 0.58 | 0.86 | 0.59 | 0.66 | 14.63 | −5.48 |
| | Val | 25.18 (27.71) | 0.54 | 0.85 | 0.77 | 0.40 | 19.67 | 9.10 |
| TP5 | Cal | 44.94 (61.29) | 0.59 | 0.84 | 0.70 | 0.51 | 39.17 | 26.67 |
| | Val | 55.19 (78.69) | 0.65 | 0.86 | 0.69 | 0.53 | 47.21 | 29.86 |
| TP6 | Cal | 28.26 (30.95) | 0.59 | 0.87 | 0.66 | 0.57 | 20.89 | 8.63 |
| | Val | 36.03 (48.97) | 0.60 | 0.82 | 0.69 | 0.52 | 34.50 | 26.43 |

() indicate observed daily mean streamflow.

Figure 7 shows hydrograph comparisons between the calibrated and validated simulation results using SCO4 at all calibration target points (TP1–TP6). The results indicate adequate calibration and validation performance over the simulation and validation period. The timing of peak flows and the magnitude of peaks match well between the simulated and observed flows at TPs 1, 2, 3, and 4 during the calibration period. However, the magnitude of peaks at TPs 5 and 6 show somewhat different results due to the large reservoir diversion nearby. The values of D between the simulated and observed streamflows at all TPs during the calibration and validation periods ranged from 0.84 to 0.95 and 0.82 to 0.95, respectively, which is a satisfactory performance as shown in Table 5.

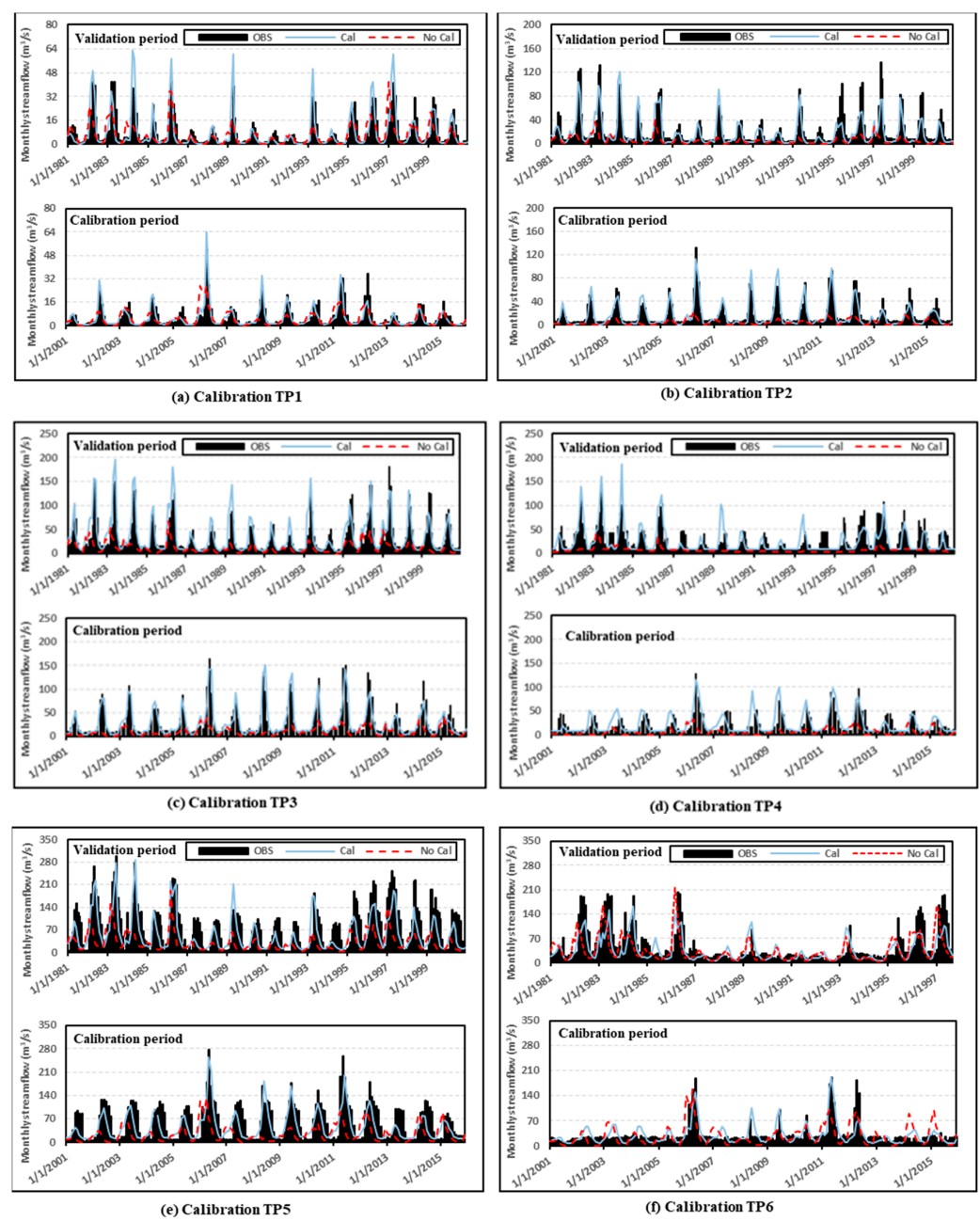

**Figure 7.** Hydrograph comparison at the calibration target points (TP1 (**a**), TP2 (**b**), TP3 (**c**), TP4 (**d**), TP5 (**e**), TP6 (**f**)) between the simulated flows (before and after calibration) and the observed flows for the validation period (1981–2000) and calibration period (2001–2015) using HSPF in sLCS.

**Table 5.** General model performance rating for recommended statistics at monthly time steps.

| Performance Rating | R²[1] | RSR[2] | NSE[2] | PBIAS[1] |
|---|---|---|---|---|
| Very good | $0.85 < R^2$ | $0.00 \leq RSR \leq 0.50$ | $0.75 < NSE$ | $PBIAS < \pm 10$ |
| Good | $0.75 < R^2 \leq 0.85$ | $0.50 < RSR \leq 0.60$ | $0.65 < NSE\ 0.85$ | $\pm 10 < PBIAS\ \pm 15$ |
| Fair | $0.65 < R^2 \leq 0.75$ | $0.60 < RSR \leq 0.7$ | $0.50 < NSE\ 0.65$ | $\pm 15 < PBIAS\ \pm 25$ |
| Poor | $R^2 \leq 0.65$ | $RSR > 0.7$ | $NSE \leq 0.50$ | $PBIAS > \pm 25$ |

Note that the values of 1 and 2 are adopted from [57] and [58], respectively.

### 4.4. Results of Streamflow Analysis Using SCO4

Figure 8a shows the time series of the simulated annual streamflow using SCO4 for 1981–2015 at the watershed outlet. In general, annual streamflow shows a negative trend with minimum, mean, and maximum flow of 12.24 m$^3$/s, 31.89 m$^3$/s, 59.14 m$^3$/s, respectively, while a positive trend is also observed when subsets of annual streamflow are used (Figure 8b). Thus, the simulation periods are divided into five subsets for visual inspection with 1981–1986 (Sub 1), 1987–1994 (Sub 2), 1995–1998 (Sub 3), 1999–005 (Sub 4), and 2006–2015 (Sub 5). The trend lines for the respective time window are then generated as shown in Figure 8c.

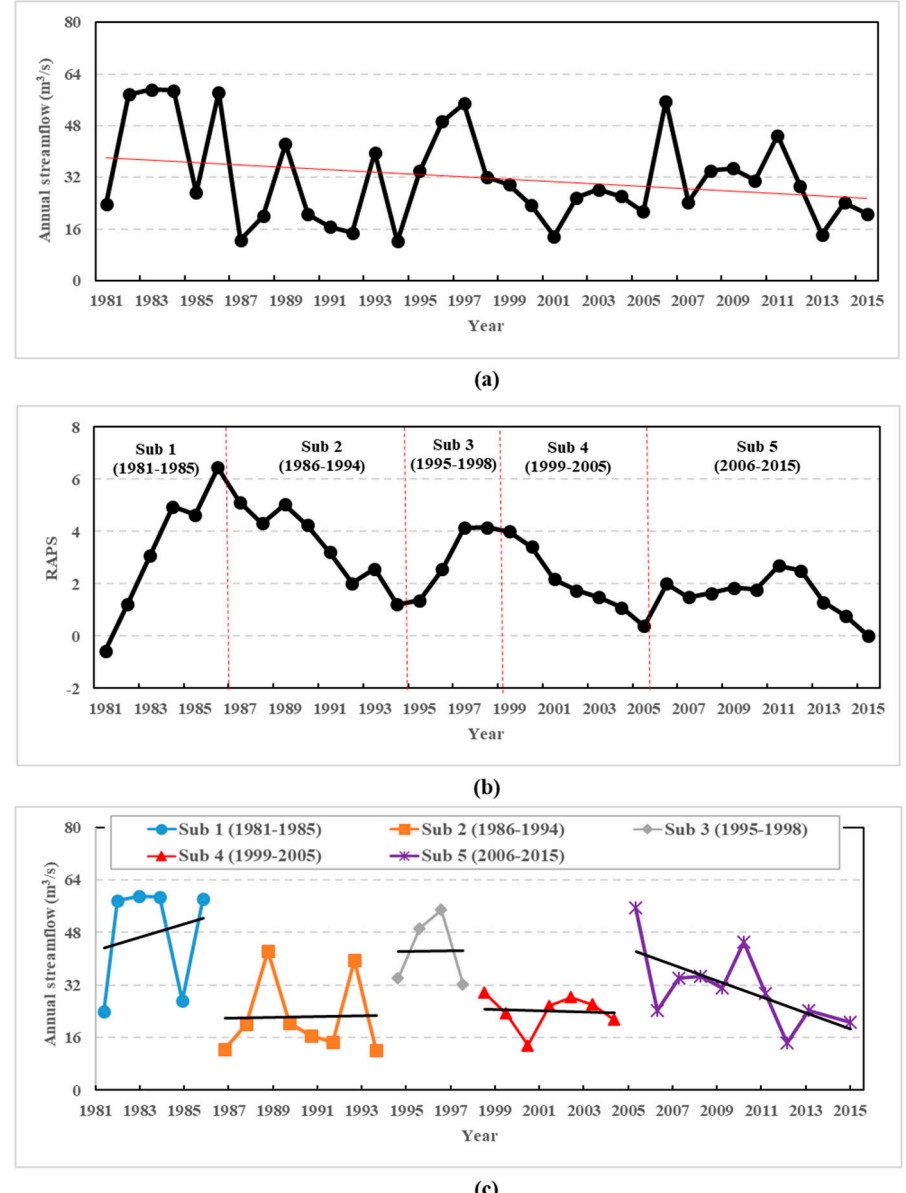

**Figure 8.** Time series of (**a**) annual streamflow, (**b**) the Rescaled Adjusted Partial Sums (RAPS) for annual streamflow, and (**c**) annual streamflow with trend lines for the subsets of the simulation periods, 1981–2015.

Our simulation model results show reliable model simulation performance based on the evaluation criteria. However, it is difficult to determine exact model performance since analyzed statistical criteria provide different performance ratings from very good to fair performance depending on the selected criteria. Therefore, the integrated criteria of model performance, such as the ideal point error (IPE)

metric [59–62] or the standardized ranking performance index (sRPI) [63] could be additional measures to evaluate more robust model performance in a future study.

## 5. Conclusions

Computer parallelism is a useful tool to reduce the computation time. Hydrologic model calibration in parallel computing can provide various opportunities to improve model performance, yet its qualitative performance has not been reported in the hydrology community. To quantify those performances in hydrological simulations, four different calibration schemes are employed and BEOPEST is used as a tool to parallelize the HSPF model in sLCS. Performance measures of parallelism, including program run time (PT), percentage of time reduction (PP), parallel speedup (PS), and parallel efficiency (PE) are used along with other evaluation criteria for hydrological simulations, which include: index of agreement (D), coefficient of determination (R$^2$), Nash–Sutcliffe efficiency (NSE), root mean square error (RMSE), RMSE-observations standard deviation ration (RSR), and percentage of bias (PBIAS).

The results show that total PT during calibration is tremendously reduced by 76%, 89%, 89%, and 90% from SCO 1, SCO 2, SCO 3, and SCO 4, respectively, when 14 cores are used instead of a single core. Additionally, SCO4 outperforms others based on its performance measures described early. As such, SCO4 is used for further analysis to improve model performance for all calibration target points (TP1–TP6). After model calibration based on SCO4, annual streamflow trends are also observed for the interested reader using RAPS for the subsets (Sub 1 thru Sub 5) of the annual streamflow in different time windows.

We can conclude that computer parallelism, with many parameters at multiple sub-watersheds, will benefit hydrologists for improvement of hydrological simulations. In addition, the proposed method will provide great potential for reliable water quality and quantity simulations when large reservoir and irrigation components are fully incorporated into HSPF. Therefore, the proposed case study is a good example for hydrologists to apply computer parallelism using sLCS to their own applications, including but not limited to streamflow, physical and conceptual hydrologic, and environmental simulations in a changing global environment.

**Author Contributions:** J.J.K. applied HSPF model in computer parallelism, coding, and analysis; and is the primary author on the manuscript. J.R. proposed the study and contributed to conceptualizing the project, interpreting the processes in general as J.J.K.'s advisor.

**Funding:** This research is supported partially by the National Institute of Food and Agriculture, U.S. Department of Agriculture (USDA), under ID01507 and the Idaho State Board of Education (ISBOE) through IGEM program. Any opinions, findings, conclusions, or recommendations expressed in this publication are those of the authors and do not necessarily reflect the view of USDA and ISBOE

**Conflicts of Interest:** The authors declare no conflict of interest.

## Appendix A

$$D = 1.0 - \frac{\sum_{i=1}^{N}(Q_{Oi} - Q_{Si})^2}{\sum_{i=1}^{N}\left(\left|Q_{Si} - \overline{Q}_{Oi}\right| + \left|Q_{Oi} - \overline{Q}_{Oi}\right|\right)^2} \tag{A1}$$

D = insensitivity of correlation-based measure to variances (Wilmott, 1984).

$$R^2 = \left(\frac{N \times \sum_{i=1}^{N}(Q_{Oi} \times Q_{Si}) - \left(\sum_{i=1}^{N} Q_{Oi}\right) \times \left(\sum_{i=1}^{N} Q_{Si}\right)}{\sqrt{N \times \left(\sum_{i=1}^{N} Q_{Oi}^2\right) - \left(\sum_{i=1}^{N} Q_{O1}\right)^2} \times \sqrt{N \times \left(\sum_{i=1}^{N} Q_{Si}^2\right) - \left(\sum_{i=1}^{N} Q_{S1}\right)^2}}\right)^2 \tag{A2}$$

$$NSE = 1.0 - \left[\frac{\sum_{i=1}^{N}(Q_{Oi} - Q_{Si})^2}{\sum_{i=1}^{N}\left(Q_{Oi} - \overline{Q}_{Oi}\right)^2}\right] \tag{A3}$$

$$RMSE = \left[\frac{1}{N}\sum_{i=1}^{N}(Q_{Oi} - Q_{si})^2\right]^{0.5} \tag{A4}$$

$$RSR = \frac{RMSE}{STDEV_{obs}} = \frac{\sqrt{\sum_{i=1}^{N}\left(Q_{Qi} - Q_{si}\right)^2}}{\sqrt{\sum_{i=1}^{N}\left(Q_{Oi} - \overline{Q}_{Oi}\right)^2}} \tag{A5}$$

$$PBIAS = \frac{\sum_{i=1}^{N}(Q_{Oi} - Q_{Si})}{\sum_{i=1}^{N}Q_{Oi}} \times 100 \tag{A6}$$

where, $Q_{Oi}$ and $Q_{Si}$ are observed and simulated streamflow at time step, respectively. $\overline{Q}_{Oi}$ and $\overline{Q}_{Si}$ are the mean observed and simulated streamflow for the simulation period. N is total number of values within the simulation period.

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
