# Peer review of "Quantifying the Performances of the Semi-Distributed Hydrologic Model in Parallel Computing—A Case Study"

_water, doi:10.3390/w11040823_

Round 1

Reviewer 1 Report

Manuscript has high scientific and practical value. I would like to recommend some guidelines, which (in my opinion) must be included, and provided analysis must be changed. I got some major concerns.

1) First of them is presented scale, i.e. time frame for analysis. I can not figured it out if presented method is appropriate for smaller time units, for example in one year scale or even smaller. Please, can you explain this?

2) From the Figure 7, I can not conclude if river has stream characteristics or the flow is mostly steady. Is presented methodology applicable for both cases? I would like to read explanation for situation when we have stream hydrograph (with so called delayed lag, when regular flow is low, and one or two days after precipitation peak of the hydrograph is rapidly growing) and hydrograph when we don't have the above mentioned situation. Also, I suggest to put legend on each particular figure; not only on the first left.

3) Authors provide testing procedure of the calculated values, i.e. of values obtained with simulation. If I have understood correctly, simulation model generated yearly time serie(s) of the flow. I would like if authors may apply Rescaled Adjusted Partial Sums (RAPS) method on obtained time series. This analysis would definitelly show if there are some fluctuations or subseries in obtained time serie(s). I insist on this, because RAPS analysis will definitelly visualize/indicate if there are some existence of trends, sudden decrease or increasing of the values, grouping of data, irregularities behaviors, periodic repetitions, etc.

Author Response

Thanks for reviewing our manuscript. Our response is enclosed in pdf file. 

Reviewer 2 Report

The paper presents a quantification of parallel computing performance in the use of hydrological models. The subject is of great interest for hydrologists that suffer from longtime simulations. I believe that the paper can be published after some minor additions regarding the section 3.6.2.

A basic problem in hydrological simulations is not only the high simulation time, but also the evaluation of many simulation scenarios using statistical criteria in order to choose the best calibration scenario. The selected case study made it look easy but in reality it does not.

If we introduce highly advanced parallel programming in hydrological simulations, we will face the problem of multiple solutions with high statistical perfrormance capturing different aspects of the hydrograph. I would suggest to add some lines regarding this problem to a discussion section, which is missing from the manuscript. Some suggestions for solving this problem are the methods of  IPE index (Elshorbagy et al., 2010; Domínguez et al., 2011; Dawson et al.,

2012) and RPI index (Aschonitis et al. 2019).

https://doi.org/10.5194/hess-14-1931-2010

https://doi.org/10.5194/hess-14-1943-2010

https://doi.org/10.5194/hess-16-3049-2012

https://doi.org/10.2166/hydro.2010.116

https://doi.org/10.1016/j.envsoft.2019.01.005

Author Response

(The authors gave the same response as above.)

Round 2

Reviewer 1 Report

In this iteration, paper is improved. Despite of this, I am recommending major revision. There are many reasons for this.

-author have describe and apply RAPS method. this should be mentioned in Abstract.

-title of figure 6 should be under mentioned figure.

-at figure 7d, please remove last ''n'' from the title of figure (calibrationn).

-subchapter 4.4. new part of the text should be aligned from both margins.

-RAPS analysis is not provided correctly. Before figure 8, original time serie must be provided. Presentation of RAPS shows any new subserie. You can not perform analysis of the value pattern by on such way. Figure 8 shows time periods where original time serie should be divided. So, correct order of the figures should be: Figure 8) original time serie of the flow. Figure 9) existing figure 8. Figure 10. New subseries. Discussion provided for the existing Figure 8 (the one with RAPS), on same manner should be provided for last figure with new subseries. Please, correct Conclusion with new comments and cognitions in subchapter 4.4.

-I do not support adding of the Discussion into Conclusion. Please, separate this chapters.

Author Response

Review 1.

In this iteration, paper is improved. Despite of this, I am recommending major revision. There are many reasons for this.

1.       Author have describe and apply RAPS method. This should be mentioned in Abstract.

-          As per reviewer’s request, we mentioned RAPS in the revised Abstract below.

The research features how parallel computing can advance hydrological performances associated with different calibration schemes (SCOs). The result shows that parallel computing can save execution time up to 90%, while simulation improvement is achieved by 81%. Basic statistics, including 1) index of agreement (D), 2) coefficient of determination (R2), 3) root mean square error (RMSE), and 4) percentage of bias (PBIAS) are used to evaluate simulation performances after model calibration in computer parallelism. Once the best calibration scheme is selected, additional efforts are made to improve model performances at the selected calibration target points, while the Rescaled Adjusted Partial Sums (RAPS) is used to evaluate the trend in annual streamflow. The qualitative result of execution time reduction by 86% in average indicate that parallel computing is another avenue to advance hydrologic simulations in the urban-rural interface, such as the Boise River Watershed, Idaho. Therefore, the research will provide useful insights for hydrologists to design and set up their own hydrological modeling exercises in the cost-effective parallel computing described in this case study.”

2.       Title of figure 6 should be under mentioned figure.

-          Thank you. it is fixed now.

3.       At figure 7d, please remove last “n” from the title of figure.

-          Thank you. We fixed it.

4.       Subchapter 4.4 new part of the text should be aligned from both margins.

-          Thank you.

5.       RAPS analysis is not provided correctly. Before figure 8, original time serie must be provided. Presentation of RAPS shows any new subserie. You can not perform analysis of the value pattern by on such way. Figure 8 shows time periods where original time serie should be divided. So, correct order of the figures should be: Figure 8) original time serie of the flow. Figure 9) existing figure 8. Figure 10. New subseries. Discussion provided for the existing Figure 8 (the one with RAPS), on same manner should be provided for last figure with new subseries. Please, correct Conclusion with new comments and cognitions in subchapter 4.4.

-          As per reviewer’s request, we included figures for original time series, RAPA time series, and the defined subsets time series in result section. Please see the revised manuscript. Also note that the goal of this paper is to more highlight computer parallelism rather than simulations in details.

6.       I do not support adding of the Discussion into Conclusion. Please, separate this chapters.

-          As per reviewer’s request, we reorganized the manuscript. Thank you.  

Reviewer 2 Report

The paper can be accepted after correcting the style of references in the text. The citations in the text should appear using [1] based on the order of appearance, while they should be ordered at the end of the reference list. The authors should take a look some puplished articles of the journal in order to see how to do it.

Author Response

Review 2.

The paper can be accepted after correcting the style of reference in the text. The citations in the text should appear using [1] based on the order of appearance, while they should be ordered at the end of the reference list. The authors should take a look some punished articles of the journal in order to see how to do it.

-          As per reviewer’s request, we have revised the manuscript accordingly. Thanks for the suggestion.